# Quasi-Periodic and Fractal Polymers: Energy Structure and Carrier Transfer

**DOI:** 10.3390/ma12132177

**Published:** 2019-07-06

**Authors:** Marilena Mantela, Konstantinos Lambropoulos, Marina Theodorakou, Constantinos Simserides

**Affiliations:** Department of Physics, National and Kapodistrian University of Athens, Panepistimiopolis, Zografos, GR-15784 Athens, Greece

**Keywords:** aperiodic, quasiperiodic, fractal, energy structure, charge transfer

## Abstract

We study the energy structure and the coherent transfer of an extra electron or hole along aperiodic polymers made of *N* monomers, with fixed boundaries, using B-DNA as our prototype system. We use a Tight-Binding wire model, where a site is a monomer (e.g., in DNA, a base pair). We consider quasi-periodic (Fibonacci, Thue–Morse, Double-Period, Rudin–Shapiro) and fractal (Cantor Set, Asymmetric Cantor Set) polymers made of the same monomer (I polymers) or made of different monomers (D polymers). For all types of such polymers, we calculate the highest occupied molecular orbital (HOMO) eigenspectrum and the lowest unoccupied molecular orbital (LUMO) eigenspectrum, the HOMO–LUMO gap and the density of states. We examine the mean over time probability to find the carrier at each monomer, the frequency content of carrier transfer (Fourier spectra, *weighted mean frequency* of each monomer, *total weighted mean frequency* of the polymer), and the *pure* mean transfer rate *k*. Our results reveal that there is a correspondence between the degree of structural complexity and the transfer properties. I polymers are more favorable for charge transfer than D polymers. We compare k(N) of quasi-periodic and fractal sequences with that of periodic sequences (including homopolymers) as well as with randomly shuffled sequences. Finally, we discuss aspects of experimental results on charge transfer rates in DNA with respect to our coherent pure mean transfer rates.

## 1. Introduction

Today, the electronic structure of biological molecules (e.g., proteins, enzymes, peptides and nucleic acids (DNA, RNA)) and their charge transfer and transport properties attract considerable interest among the physical, chemical, biological and medical communities, as well as a broad spectrum of interdisciplinary scientists and engineers [1,2,3,4,5,6]. DNA plays a fundamental role in genetics and molecular biology since its sequence of bases, adenine (A), guanine (G), cytosine (C), and thymine (T) contains the genetic code of living organisms. The base-pair stack of the DNA double helix creates a nearly one-dimensional π-pathway that favors charge transfer and transport. Although it is not rare to see or hear the terms *transport* and *transfer* indiscriminately, if we want to be more precise, *transport* means that the system under investigation is held between electrodes and that a voltage is applied between these electrodes, while the term *transfer* means that a carrier, created (e.g., by oxidation or reduction) or injected at a specific place, moves to a more favorable location, without the application of external voltage. Charge transfer through DNA plays a central role in DNA damage and repair [2,7,8], so it may be a critical issue in carcinogenesis and mutagenesis [9,10]. For example, the rapid hole migration from other bases to guanine is connected to the fact that direct strand breaks occur preferentially at guanines [9]. Furthermore, it might be an indicator of discrimination between pathogenic and non-pathogenic mutations at an early stage [11].

Charge movement is usually ascribed to two types of mechanisms [12,13]: (i) *incoherent* or *thermal* hopping between nearest neighboring or more distant sites and (ii) *coherent* hopping or *tunneling* or *superexchange*. The term *tunneling* implies quantum mechanical tunneling, between two sites, e.g., the carrier donor and the carrier acceptor, through a bridge. The term *superexchange*, not to be confused with the similar term in magnetism, emanates from the distant interaction between the two sites, e.g., the donor and the acceptor, through a bridge. However, we have shown systematically [14,15,16,17,18] that, in the coherent regime, all sites contribute with finite occupation probabilities, although those with adequate on-site energies, for the initial placement of the carrier in the sequence, are more favored. This conclusion holds both for the wire model (where the site is a base pair) and for the extended ladder model (where the site is a base) that we have used so far. The coherent mechanism is expected to dominate carrier movement in the low temperature regime. In natural DNA, it is more likely that a hole will be created at a guanine which has the highest HOMO (highest occupied molecular orbital) of all bases [19] and an electron will be created at a thymine which has the lowest LUMO (lowest unoccupied molecular orbital) of all bases [19]. However, coherently, if e.g., the hole is initially created or injected at an adenine, charge transfer will mainly be accomplished through adenines and similarly for other initial conditions [16]. Typically, in *coherent* transfer, charge is never exactly localized, but there is a mean over time occupation probability to find it at each site, the carrier does not exchange energy with the environment during its transfer and this way it can travel short distances; strictly quantum mechanically, just a percentage of the carrier reaches the last site.

Typically, in *thermal* hopping, charge is localized, the carrier exchanges energy with the environment during its transfer and this way it can travel far longer than via the coherent mechanism. If $d_{0}$ is a typical nearest neighbor distance, e.g., 3.4 Å, and two sites stand off Δr having on-site energy difference ΔE, then, maybe one could presume an equation like $k=k_{0}\exp\left(-\Delta E/k_{B}T\right)\exp\left(-\Delta r/d_{0}\right)$—or a similar one with other mathematical form—to qualitatively describe thermal hopping [20,21,22,23,24,25,26,27,28,29,30,31].

In the present work, we take B-DNA as a prototype system because, apart from its biological and nanoscientific importance, it has a rather long persistence length of around 50 nm or 150 base pairs [32]. However, there are several studies concerning charge and energy transfer in other aperiodic polymer systems [33,34,35]. We study the coherent regime, cf. Equation (Equation 12), this time for aperiodic polymers. Although unbiased coherent charge transfer in DNA nearly vanishes after 10 to 20 nm [14,15,16,18], DNA still remains a promising candidate as an electronic component in molecular electronics, e.g., as a short molecular wire or a nanocircuit element [36,37]. Favoring geometries and base-pair sequences have still to be explored, e.g., incorporation of sequences serving as molecular rectifiers, use of non-natural bases or using the triplet acceptor anthraquinone for hole injection [37].

Research has recently shown that carrier movement through B-DNA can be manipulated. Using various natural and artificial nucleobases (chemical modification) with different HOMO levels, the hole transfer rate through DNA can be tuned [38]. The carrier transfer rate strongly depends on the difference between HOMO energies (for hole transfer) or LUMO energies (for electron transfer) and so it can be increased by many orders of magnitude with appropriate sequence choice [15,16,17,18]. Furthermore, structural fluctuations is another factor which influences quantum transport through DNA molecular wires [39].

We know that many factors (e.g., aqueousness, counterions, extraction process, electrodes, purity, substrate, structural fluctuations, geometry) influence carrier motion along DNA. These factors are either intrinsic or extrinsic. Here, we focus on the most important of the intrinsic factors, i.e., the effect of alternating the base-pair sequence, which affects the overlaps across the π-stack. The aim of this work is a comparative examination of the influence of base-pair sequence on charge transfer, in aperiodic sequences. Ab initio calculations [40,41,42,43,44,45,46,47,48], used to explore experimental results and the underlying mechanisms, are currently limited to short segments for computational reasons. Here, we study rather long sequences, so we employ the Tight-Binding (TB) model which allows for addressing systems of realistic length [14,15,16,49,50,51,52,53,54,55,56,57,58,59,60,61,62,63,64,65].

There are several works devoted to the study of transfer and transport in specific DNA structures using variants of the Tight-Binding method [12,15,16,50,51,63,66,67,68,69]. Here, we employ a TB wire model, where the base pairs (or monomers) are the sites of the chain, to study the spectral and charge transfer properties of deterministic aperiodic (Thue–Morse (TM), Fibonacci (F), Double-Period (DP), Rudin–Shapiro (RS), Cantor Set (CS), Asymmetric Cantor Set (ACS)) DNA segments. Th relevant parameters are the on-site energies of base pairs and the hopping integrals between successive base pairs. We have to solve a system of *N* coupled equations for the time-independent problem, and a system of *N* coupled first order differential equations for the time-dependent problem. We study HOMO and LUMO eigenspectra, HOMO–LUMO gaps and the relevant density of states (DOS) as well as the mean over time probabilities to find the carrier at each site. We are also interested in the frequency content of carrier movement; hence, we analyze the Fourier spectra of the time-dependent probability to find the carrier at each site, the weighted mean frequency of each monomer and the total weighted mean frequency of the polymer. Finally, we study the pure mean transfer rate from a certain site to another, which describes the easiness of charge transfer; it gives us a measure of how much of the carrier is transferred and also of how fast this process is. Recently, charge transport (i.e., the movement of a carrier under external bias) in aperiodic polymers has been studied in Ref. [70]. Specifically, Ref. [70] was devoted to autocorrelation functions, Lyapunov eexponents, transmission coefficients, and current–voltage curves. Here, on the contrary, we study charge transfer (i.e., the movement of a carrier created via oxidation (hole) or reduction (electron), without external bias) along aperiodic polymers.

The rest of the paper is organized as follows: in Section 2, we provide some details on the studied deterministic aperiodic sequences and we outline our notation. In Section 3, we delineate the basic theory behind the time-independent (Section 3.1) and the time-dependent (Section 3.2) problem. In Section 4, we discuss our results for polymers made of the same monomer and polymers made of different monomers. Here, for DNA, a monomer is a base pair. Finally, in Section 5, we state our conclusions. The analysis of our results is done with the equations included in Section 3. Extensive analysis of our fitting methods can be found in Refs. [14,15,16] and in the Supplemental Material of Ref. [18].

## 2. Sequences and Notation

In our prototype system, B-DNA, we mention only the base sequence of the 5${}^{\prime }$-3${}^{\prime }$ strand. For example, we denote two successive monomers by YX, meaning that the base pair X–X${}_{\mathrm{compl}}$ is separated and twisted by 3.4 Å and $36^{\circ }$, respectively, relatively to the base pair Y-Y${}_{\mathrm{compl}}$, around the B-DNA growth axis. X${}_{\mathrm{compl}}$ (Y${}_{\mathrm{compl}}$) is the complementary base of X (Y).

The deterministic aperiodic sequences considered in this work are either quasi-periodic or fractal. Such structures are generally known as binary substitutional sequences, i.e., based on a binary alphabet, like {0, 1} and generated using appropriate substitution rules.

### 2.1. Fibonacci

The Fibonacci (F) sequence is named after the Italian mathematician Leonardo Pisano (Fibonacci) who introduced it to Western European mathematics in his 1202 book Liber Abaci, in a study of the population growth of rabbits [71]. However, this sequence appears many centuries before in Indian mathematics [72]. Fibonacci considers the growth of an idealized rabbit population, assuming that a single newly born pair of rabbits (N) are put in a field, and rabbits are able to mate at the age of one month so that, at the end of its second month, a mature pair (M) can produce another pair of rabbits. Rabbits never die and a mating pair always produces one new pair every month from the second month on. The puzzle that Fibonacci posed was: how many pairs will exist in one year? The collection of every month’s population is: $F_{0}=$ N, $F_{1}=$ M, $F_{2}=$ MN, $F_{3}=$ MNM, $F_{4}=$ MNMMN, etc. Using e.g., the two-letter alphabet {G, A}, we can define the Fibonacci generation $F_{g}$ by the substitution rules A → G, G → GA, starting with $F_{0}=$ A. Hence, $F_{0}=$ A, $F_{1}=$ G, $F_{2}=$ GA, $F_{3}=$ GAG, $F_{4}=$ GAGGA, etc. If $N_{g}$ is the Fibonacci number of generation *g*, and we set $N_{0}=N_{1}=1$, the recurrence relation $N_{g}=N_{g-1}+N_{g-2}$ produces the number sequence 1, 1, 2, 3, 5, 8, 13, 21, 34, …

### 2.2. Thue–Morse

The Thue–Morse (TM) or Prouhet–Thue–Morse sequence was first studied by Eugène Prouhet in 1851, who applied it to number theory [73]. The systematic study was left to Axel Thue who, in 1906, applied it on his study of words combinatorics [74]. The most important contribution to the sequence was made in 1921 by Marston Morse in the context of differential geometry and topological dynamics [75], which brought the sequence to worldwide attention. In its simplest form, the TM sequence can be defined by the recursive relations $S_{n}=\left\{S_{n-1}S_{n-1}^{+}\right\}$ and $S_{n}^{+}=\left\{S_{n-1}^{+}S_{n-1}\right\}$ (for n≥1), with $S_{0}=0$ and $S_{0}^{+}=1$ [76]. Using e.g., the two-letter alphabet {G, A} we can build up the sequence using the substitution rules G→GA and A→AG. Hence, ${\mathrm{TM}}_{0}=$ G, ${\mathrm{TM}}_{1}=$ GA, ${\mathrm{TM}}_{2}=$ GAAG, ${\mathrm{TM}}_{3}=$ GAAGAGGA, etc.

### 2.3. Double-Period

The double-period (DP) sequence has its origin in the study of system dynamics and laser applications to nonlinear optical fibers [77]. It is closely connected with the TM sequence: the *n*-th stage is $S_{n}=\left\{S_{n-1}S_{n-1}^{+}\right\}$ and $S_{n}^{+}=\left\{S_{n-1}S_{n-1}\right\}$ (for n≥1), with $S_{0}=0$ and $S_{0}^{+}=1$. Using e.g., the two-letter alphabet {G, A}, we can define the *n*-th generation by the substitution rules G→GA, A→GG. Hence, starting with ${\mathrm{DP}}_{0}=$ G, then ${\mathrm{DP}}_{1}=$ GA, ${\mathrm{DP}}_{2}=$ GAGG, ${\mathrm{DP}}_{3}=$ GAGGGAGA, etc.

### 2.4. Rudin–Shapiro

The Rudin–Shapiro (RS) aka Golay–Rudin–Shapiro sequence is named after Marcel Golay, Walter Rudin and Harold S. Shapiro, who independently investigated its properties [78,79,80]. It is generated starting with +1, +1 and employing the rules:$$\begin{array}{c} +1,+1\to +1,+1,+1,-1,\\ +1,-1\to +1,+1,-1,+1,\\ -1,+1\to -1,-1,+1,-1,\\ -1,-1\to -1,-1,-1,+1. \end{array}$$

Using e.g., the two-letter alphabet {G, A} and employing the inflation rule: GG→GGGA, GA→GGAG, AG→AAGA, AA→AAAG, the first generations are ${\mathrm{RS}}_{1}=$ GG, ${\mathrm{RS}}_{2}=$ GGGA, ${\mathrm{RS}}_{3}=$ GGGAGGAG, etc.

### 2.5. Cantor Set

The Cantor Set (CS), introduced by mathematician Georg Cantor, is one of the most well-known deterministic fractals [81]. It is built by splitting a straight line segment in three, removing the middle third, then removing the middle third of each of the two new straight line segments and the process is repeated ad infinitum. Using, e.g., the two-letter alphabet {G, A} and the substitution rules G→GAG, A→AAA, we can define the *n*-th generation (n= 0, 1, 2, …) as follows: ${\mathrm{CS}}_{0}=$ G, ${\mathrm{CS}}_{1}=$ GAG, ${\mathrm{CS}}_{2}=$ GAGAAAGAG, etc.

### 2.6. Asymmetric Cantor Set

The Asymmetric Cantor Set (ACS) is built by splitting a straight line segment in four, removing the second quarter, then removing the second quarter of each of the three new straight line segments and the process is repeated ad infinitum. Using, e.g., the two-letter alphabet {G, A} and the substitution rules G→GAGG, A→AAAA, we can define the *n*-th generation (n= 0, 1, 2, …) as follows: ${\mathrm{ACS}}_{0}=$ G, ${\mathrm{ACS}}_{1}=$ GAGG, ${\mathrm{ACS}}_{2}=$ GAGGAAAAGAGGGAGG, etc.

One could think of many types of aperiodic polymers, some of which are shown synoptically in Table 1. We just give an example of each type, e.g., for Fibonacci I sequences, we give the example G, C, CG, CGC, CGCCG, CGCCGCGC, …, but there are obviously other similar sequences e.g., C, G, GC, GCG, GCGGC, GCGGCGCG, …, A, T, TA, TAT, TATTA, TATTATAT, …, T, A, AT, ATA, ATAAT, ATAATATA, …

## 3. Theory

In this article, we use a simple wire model, where the site is a monomer (e.g., in DNA, a base pair). We call μ the monomer index, μ=1,2,…,N. We assume that the state or movement of an extra hole or electron can be expressed through the monomer HOMOs or LUMOs, respectively, cf. Equations (Equation 2) and (Equation 8) below.

### 3.1. Stationary States—Time-Independent Problem

The TB *wire* model Hamiltonian can be written as
(1)$${\hat{H}}_{W}=\sum_{\mu =1}^{N}E_{\mu }|\mu \rangle \langle \mu |+\left(\sum_{\mu =1}^{N-1}t_{\mu ,\mu +1}|\mu \rangle \langle \mu +1|+h.c.\right).$$
$E_{\mu }$ is the on-site energy of the μ-th monomer, and $t_{\mu ,\lambda }=t_{\lambda ,\mu }^{*}$ is the hopping integral between monomers μ and λ. The state of a polymer can be expressed as
(2)$$|P\rangle =\sum_{\mu =1}^{N}v_{\mu }|\mu \rangle .$$

Substituting Equations (Equation 1) and (Equation 2) to the time-independent Schrödinger equation
(3)$$\hat{H}|P\rangle =E|P\rangle ,$$
we arrive to a system of *N* coupled equations
(4)$$E_{\mu }v_{\mu }+t_{\mu ,\mu +1}v_{\mu +1}+t_{\mu ,\mu -1}v_{\mu -1}=Ev_{\mu },$$
which is equivalent to the eigenvalue-eigenvector problem
(5)$$H\vec{v}=E\vec{v},$$
where H is the Hamiltonian matrix of order *N*, composed of the TB parameters $E_{\mu }$ and $t_{\mu ,\lambda }$, and $\vec{v}$ is the vector matrix composed of the coefficients $v_{\mu }$ (which can be chosen to be real). The diagonalization of H leads to the determination of the eigenenergy spectrum (*eigenspectrum*), $\left\{E_{k}\right\}$, k=1,2,…,N, for which we suppose that $E_{1}<E_{2}<\cdots <E_{N}$, as well as to the determination of the occupation probabilities for each eigenstate, $|v_{\mu k}{|}^{2}$, where $v_{\mu k}$ is the μ-th component of the *k*-th eigenvector. $\left\{v_{\mu k}\right\}$ are normalized, and their linear independence is checked in all cases.

Having determined the eigenspectrum, we can compute the density of states (DOS), generally given by
(6)$$g\left(E\right)=\sum_{k=1}^{N}\delta \left(E-E_{k}\right).$$

Changing the view of a polymer from one (e.g., top) to the other (e.g., bottom) side of the growth axis, reflects the Hamiltonian matrix H of the polymer on its main antidiagonal. This reflected Hamiltonian, $H^{\mathrm{equiv}}$, describes the *equivalent polymer* [16]. H and $H^{\mathrm{equiv}}$ are connected by the similarity transformation $H^{\mathrm{equiv}}=L^{-1}\mathrm{HL}$, where $L(=L^{-1})$ is the unit antidiagonal matrix of order *N*. Therefore, H and $H^{\mathrm{equiv}}$ have identical eigenspectra (hence the equivalent polymers’ DOS is identical) and their eigenvectors are connected by $v_{\mu k}=v_{(N-\mu +1)k}^{\mathrm{equiv}}$. Generally,
(7)$$\mathrm{equiv}(\mathrm{YX}\ldots Z)=Z_{\mathrm{compl}}\ldots Y_{\mathrm{compl}}X_{\mathrm{compl}}.$$

### 3.2. Time-Dependent Problem

To describe the spatiotemporal evolution of an extra carrier (hole/electron), inserted or created (e.g., by oxidation/reduction) at a particular monomer of the polymer, we consider the state of the polymer as
(8)$$|P(t)\rangle =\sum_{\mu =1}^{N}C_{\mu }\left(t\right)|\mu \rangle ,$$
where $|C_{\mu }\left(t\right){|}^{2}$ is the probability to find the carrier at the μ-th monomer at time *t*. Substituting Equations (Equation 1) and (Equation 8) in the time-dependent Schrödinger equation
(9)$$i\hbar \frac{\partial }{\partial t}|P(t)\rangle =\hat{H}|P(t)\rangle ,$$
we arrive at a system of *N* coupled differential equations
(10)$$i\hbar \frac{dC_{\mu }}{dt}=E_{\mu }C_{\mu }+t_{\mu ,\mu +1}C_{\mu +1}+t_{\mu ,\mu -1}C_{\mu -1}.$$

Equation (Equation 10) is equivalent to a 1st order matrix differential equation of the form
(11)$$\dot{\vec{C}}\left(t\right)=-\frac{i}{\hbar }H\vec{C}\left(t\right),$$
where $\vec{C}\left(t\right)$ is a vector matrix composed of the coefficients $C_{\mu }\left(t\right),\;\;\mu =1,2,\ldots ,N$. Equation (Equation 11) can be solved with the eigenvalue method, i.e., by looking for solutions of the form $\vec{C}\left(t\right)=\vec{v}e^{-\frac{i}{\hbar }Et}\Rightarrow \dot{\vec{C}}\left(t\right)=-\frac{i}{\hbar }E\vec{v}e^{-\frac{i}{\hbar }Et}$. Hence, Equation (Equation 11) leads to the eigenvalue problem of Equation (Equation 5), that is, $H\vec{v}=E\vec{v}$. Having determined the eigenvalues and eigenvectors of H, the general solution of Equation (Equation 11) is
(12)$$\vec{C}\left(t\right)=\sum_{k=1}^{N}c_{k}{\vec{v}}_{k}e^{-\frac{i}{\hbar }E_{k}t}.$$

In other words, the coefficients $C_{\mu }\left(t\right),\mu =1,2,\ldots ,N$ are given by a superposition of the time evolution of the stationary states with time-independent coefficients $c_{k}$. Hence, this is a coherent phenomenon. The coefficients $c_{k}$ are determined from the initial conditions. In particular, if we define the N×N eigenvector matrix v, with elements $v_{\mu k}$, then it can be shown that the vector matrix $\vec{c}$, composed of the coefficients $c_{k},\;\;k=1,2,\ldots ,N$, is given by the expression
(13)$$\vec{c}=v^{T}\vec{C}\left(0\right).$$

Suppose that initially the extra carrier is placed at the λ-th monomer, i.e., $C_{\lambda }\left(0\right)=1$, $C_{\mu }\left(0\right)=0,$∀μ≠λ. Then,
(14)$$\vec{c}={\left[\begin{array}{c} v_{\lambda 1}\ldots v_{\lambda k}\ldots v_{\lambda N} \end{array}\right]}^{T}.$$

In other words, the coefficients $c_{k}$ are given by the row of the eigenvector matrix which corresponds to the monomer the carrier is initially placed at. In this work, we choose λ=1, i.e., we initially place the carrier at the first monomer. From Equation (Equation 12), it follows that the probability to find the extra carrier at the μ-th monomer is
(15)$$|C_{\mu }\left(t\right){|}^{2}=\sum_{k=1}^{N}c_{k}^{2}v_{\mu k}^{2}+2\sum_{k=1}^{N}\sum_{\begin{array}{c} k^{\prime }=1\\ k^{\prime }<k \end{array}}^{N}c_{k}c_{k^{\prime }}v_{\mu k}v_{\mu k^{\prime }}\cos\left(2\pi f_{kk^{\prime }}t\right),$$
(16)$$f_{kk^{\prime }}=\frac{1}{T_{kk^{\prime }}}=\frac{E_{k}-E_{k^{\prime }}}{h},\;\forall k>k^{\prime },$$
are the frequencies ($f_{kk^{\prime }}$) or periods ($T_{kk^{\prime }}$) involved in charge transfer. If *m* is the number of discrete eigenenergies, then the number of different $f_{kk^{\prime }}$ or $T_{kk^{\prime }}$ involved in carrier transfer is S=m2=m!2!(m−2)!=m(m−1)2. If there are no degenerate eigenenergies (which holds for all cases studied here, but e.g., does not hold for *cyclic* homopolymers [15]), then m=N. If eigenenergies are symmetric relative to some central value, then *S* decreases (there exist degenerate $f_{kk^{\prime }}$ or $T_{kk^{\prime }}$). Specifically, in that case, $S=\frac{m^{2}}{4}$, for even *m* and $S=\frac{m^{2}-1}{4}$ for odd *m*.

From Equation (Equation 15), in the absence of degeneracy and for real $c_{k}$, $v_{\mu k}$, it follows that the mean over time probability to find the extra carrier at the μ-th monomer is
(17)$$\langle |C_{\mu }\left(t\right){|}^{2}\rangle =\sum_{k=1}^{N}c_{k}^{2}v_{\mu k}^{2}.$$

Furthermore, from Equation (Equation 15), it can be shown that the one-sided Fourier amplitude spectrum that corresponds to the probability $|C_{\mu }\left(t\right){|}^{2}$ is given by
(18)$$|F_{\mu }\left(f\right)|=\sum_{k=1}^{N}c_{k}^{2}v_{\mu k}^{2}\delta \left(f\right)+2\sum_{k=1}^{N}\sum_{\begin{array}{c} k^{\prime }=1\\ k^{\prime }<k \end{array}}^{N}\left|c_{k}c_{k^{\prime }}v_{\mu k}v_{\mu k^{\prime }}\right|\delta \left(f-f_{kk^{\prime }}\right).$$

Hence, the Fourier amplitude of frequency $f_{kk^{\prime }}$ is $2|c_{k}v_{\mu k}c_{k^{\prime }}v_{\mu k^{\prime }}|$. We can further define the *weighted mean frequency* (WMF) of monomer μ as
(19)$$f_{WM}^{\mu }=\frac{\sum_{k=1}^{N}\sum_{\begin{array}{c} k^{\prime }=1\\ k^{\prime }<k \end{array}}^{N}\left|c_{k}v_{\mu k}c_{k^{\prime }}v_{\mu k^{\prime }}\right|f_{kk^{\prime }}}{\sum_{k=1}^{N}\sum_{\begin{array}{c} k^{\prime }=1\\ k^{\prime }<k \end{array}}^{N}\left|c_{k}v_{\mu k}c_{k^{\prime }}v_{\mu k^{\prime }}\right|}.$$

WMF expresses the mean frequency content of the extra carrier oscillation at monomer μ. Having determined the WMF for all monomers, we can now obtain a measure of the overall frequency content of carrier oscillations in the polymer: Since $f_{WM}^{\mu }$ is the weighted mean frequency of monomer μ and $\langle |C_{\mu }{\left(t\right)|}^{2}\rangle$ is the mean probability of finding the extra carrier at monomer μ, we define the *total weighted mean frequency* (TWMF) as
(20)$$f_{TWM}=\sum_{\mu =1}^{N}f_{WM}^{\mu }\langle |C_{\mu }{\left(t\right)|}^{2}\rangle .$$

A quantity that evaluates simultaneously the magnitude of coherent charge transfer and the time scale of the phenomenon is the *pure* mean transfer rate [14]
(21)$$k_{\lambda \mu }=\frac{\langle |C_{\mu }\left(t\right){|}^{2}\rangle }{t_{\lambda \mu }}.$$
$t_{\lambda \mu }$ is the *mean transfer time*, i.e., having placed the carrier initially at monomer λ, the time it takes for the probability to find the extra carrier at monomer μ, $|C_{\mu }\left(t\right){|}^{2}$, to become equal to its mean value, $\langle |C_{\mu }\left(t\right){|}^{2}\rangle$, for the first time. For the *pure* mean transfer rates,
(22)$$\begin{array}{c} k_{\lambda \mu }=k_{\mu \lambda }=\\ k_{\left(N-\lambda +1)(N-\mu +1\right)}^{\mathrm{equiv}}=k_{\left(N-\mu +1)(N-\lambda +1\right)}^{\mathrm{equiv}}, \end{array}$$
where the superscript “equiv” refers to the equivalent polymer in the sense of Equation (Equation 7).

## 4. Results

In this article, the TB parameters for B-DNA are taken from Ref. [19]. The HOMO/LUMO base-pair on-site energies are [19] $E_{G-C}=$ −8.0/−4.5 eV, $E_{A-T}=$ −8.3/−4.9 eV. The hopping integrals are given in Table 2.

At this point, we mention that any sign alteration of the hopping integrals does not affect the results presented below, since the Hamiltonian matrices we deal with in the Wire Model are irreducible, symmetric and tridiagonal. This can be shown as follows: let us suppose a N×N irreducible tridiagonal Hermitian matrix T, with diagonal elements $T_{k}=a_{k}$ and non-diagonal elements $T_{\left(k,k+1\right)}=r_{k+1}e^{-i\theta_{k+1}}$, $r_{k+1}>0$, ∀k=1,…N−1. Since T is Hermitian, $T_{\left(k+1,k\right)}=r_{k+1}e^{i\theta_{k+1}}$. Now, suppose a diagonal N×N matrix D, with elements $d_{1}=1,d_{k}=d_{k-1}e^{i\theta_{k}},\forall k=2,\ldots ,N$. Then, D is unitary, and the similarity transformation $\tilde{T}=D^{-1}\mathrm{TD}$ leads to the matrix $\tilde{T}$ with diagonal elements ${\tilde{T}}_{k}=a_{k}$ and non-diagonal elements ${\tilde{T}}_{\left(k,k+1\right)}=r_{k+1}$. Hence, the tridiagonal Hermitian matrix T has the same eigenvalues with the tridiagonal real symmetric matrix $\tilde{T}$, which has positive non-diagonal entries [82]. Let us further suppose that T is real. Then, $\theta_{k}=0$ or $\theta_{k}=\pi$ depending on whether $T_{\left(k,k+1\right)}>0$ or $T_{\left(k,k+1\right)}<0$. The elements of D will be $d_{k}=d_{k-1}\left(\pm 1\right),\forall k=2,\ldots ,n$. Hence, the matrix $\tilde{T}$, which has positive entries, has the same eigenvalues with T, which differs by $\tilde{T}$ in that its off-diagonal elements have negative signs in arbitrary positions. Finally, if $\vec{v}$ is an eigenvector of T, then $D^{-1}\vec{v}$ is an eigenvector of $\tilde{T}$.

### 4.1. Eigenspectra, Density of States, Energy Gaps

In Figure 1 and Figure 2, we present the HOMO and LUMO eigenspectra, for increasing *N*, of I and D polymers, respectively, and in Figure 3 and Figure 4 the corresponding DOS. In Figure 1 and Figure 2, different colors correspond to different generations, e.g., in the Fibonacci sequence, the number of monomers in the polymer, *N*, takes the following values from generation to generation (1, 1,) 2, 3, 5, 8, 13, 21, 34, 55, 89, 144, … For both I and D polymers, we notice that in quasi-periodic polymers the DOS has rather acute subbands, while in fractal polymers the DOS is fragmented and spiky. In Figure 3 and Figure 4, for illustration purposes, the DOS has been calculated for polymers made of a very large number of monomers *N*. This value is shown in each panel. Of course, the persistence length of DNA is around 50 nm or 150 base pairs [32]. On the other hand, if we stretch and join the DNA of all chromosomes of a single cell that would give us a length of the order of a meter and would consist of billions of base pairs.

For I polymers, i.e., polymers made of identical monomers (cf. Figure 1 and Figure 3), we observe that all eigenvalues are symmetric relative to the monomer’s on-site energy (this, obviously, also holds for the DOS). This observation can be mathematically proven as follows: for *N* even, the Hamiltonian matrix of a generic I polymer is $H=E_{\mu }I+T_{GK}$, where $E_{\mu }$ is the (constant) on-site energy, I is the identity matrix and $T_{GK}$ is the Golub–Kahan matrix, containing only the non-diagonal elements of H, i.e., the HOMO or LUMO hopping integrals $t_{\mu ,\lambda }$. It can easily be shown that $T_{GK}=P^{T}\mathrm{BP}$, where P is the perfect shuffle matrix and
(23)$$B=\left(\begin{array}{cc} O & A\\ A^{T} & O \end{array}\right),\;A=\left(\begin{array}{cccc} t_{1,2} & t_{2,3} & & \\ & t_{3,4} & t_{4,5} & \\ & & \ddots & \ddots \\ & & & t_{N-1,N} \end{array}\right).$$

By performing the Singular Value Decomposition of the upper bidiagonal matrix A, i.e., by writing it as $A={\mathrm{USW}}^{T}$, we obtain
(24)$$B=J\left(\begin{array}{cc} -S & 0\\ 0 & S \end{array}\right)J^{T},\;J=\frac{1}{\sqrt{2}}\left(\begin{array}{cc} U & U\\ -W & W \end{array}\right).$$

Thus, finally,
(25)$$T_{GK}=P^{T}J\left(\begin{array}{cc} -S & 0\\ 0 & S \end{array}\right)J^{T}P.$$

Hence, the eigenvalues of $T_{GK}$ are given by the positive and negative values of the diagonal matrix S, i.e., they are symmetric around zero [83]. Hence, since, $H=E_{\mu }I+T_{GK}$, the eigenvalues of H are symmetric around $E_{\mu }$. For *N* odd, we can add a zero row and a zero column to $T_{GK}$ so that it is again of even order and follow the aforementioned procedure. Then, two degenerate trivial eigenvalues will appear apart from the symmetric ones [84]. Thus, the eigenvalues of H occur by omitting the zero row and column, hence they are symmetric around $E_{\mu }$, which is also an eigenvalue.

For D polymers, i.e., polymers made of different monomers (cf. Figure 2 and Figure 4), the eigenenergies and the DOS gather around the two monomer’s on-site energies.

The energy gap of a monomer is the difference between its LUMO and HOMO levels. The energy gap of a polymer is the difference between the lowest level of the LUMO regime and the highest level of the HOMO regime because we assume that the orbitals—one per site—which contribute to the HOMO (LUMO) band are occupied (empty), since in both possible monomers there is an even number of $p_{z}$ electrons contributing to the π stack. In Figure 5, we present the energy gaps (calculated for large *N*; cf. Figure 3 and Figure 4) and the HOMO and LUMO band limits of all aperiodic polymers examined in this work. The G-C (A-T) monomer gap is always greater than the gaps of I polymers made of G and C or A and T. D polymers have smaller HOMO–LUMO gaps than I polymers (cf. upper panel of Figure 5). Furthermore, the lower HOMO (LUMO) band limit of D polymers is always between the lower and upper HOMO (LUMO) band limit of I polymers consisted of A and T, while the upper HOMO (LUMO) band limit of D polymers is always between the lower and upper HOMO (LUMO) band limit of I polymers consisted of G and C (cf. lower panel of Figure 5).

### 4.2. Mean over Time Probabilities

The main aspects of our results for the mean over time probabilities for I and D polymers are summarized in Figure 6 and Figure 7 (where we show only two consecutive generations) and in Figure A1 and Figure A2 in Appendix A (where we show many consecutive generations), for some example cases. We suppose that the extra carrier is initially placed at the first monomer. Usually, these probabilities are distributed to monomers close to the one the carrier was initially placed.

The mean over time probabilities of finding the extra carrier at each monomer of a polymer depends on the sequence on-site energies and magnitude of hopping parameters between successive monomers. This can more easily be seen in I polymers (cf. Figure 6), where only the hopping integrals affect the energy structure. For the Thue–Morse G(C) polymers, the probabilities are palindromic for odd generation numbers. This is due to the fact that the Hamiltonian matrices of these polymers are palindromic, i.e., reading them from top left to bottom right and vice versa gives the same result [18]. This property stems directly from the sequence structure. For Cantor Set A(T) polymers, the mean over time probability for an extra hole is almost totally distributed at the four (or three for generation 1) starting monomers, regardless of *N*, while for an extra electron the probabilities are almost semi-palindromic, i.e., $\langle |C_{\mu }\left(t\right){|}^{2}\rangle =\langle |C_{N-\mu +1}\left(t\right){|}^{2}\rangle ,\mu =2,4,...,N-1$. In this case, even if the sequence structure is the same for HOMO and LUMO, the magnitude of hopping integrals has a stronger effect on the results. Another example is the Rudin–Shapiro A(T) sequence where the mean over time probability for an extra electron is almost totally distributed at the four starting monomers, regardless of *N*, while for holes it is basically distributed at monomers 1, 2, 3 and 6. Regarding the extra hole in Asymmetric Cantor C(G) polymers, the probability is much higher for monomers 1, 2, 9, 10 of every 32-monomer period. Generally, for I polymers, the mean over time probabilities are significant only rather close to the first monomer, although in some cases we observe non-negligible probabilities at more distant monomers.

Generally, for D polymers, the mean over time probabilities are almost negligible further than the first monomer. An exception is the Rudin–Shapiro A(G) sequence where the probabilities for both HOMO and LUMO are almost totally distributed at the three starting monomers of each polymer, regardless its length. Likewise, the mean over time probability for the extra electron in Cantor Set A(G) polymers is almost totally distributed at the first and third monomer of each polymer, regardless its length. An extra electron in Double-Period A(G) reaches somehow more distant monomers.

A general picture is that in the *aperiodic* polymers studied in this work the mean over time probabilities generally decline a little away from the site where the carrier was initially placed. This situation is different from the general picture in *periodic* polymers studied e.g., in Refs. [15,16,18], where, as a rule, non-negligible probabilities exist at distant—from the initial—sites.

### 4.3. Frequency Content

The frequencies involved in charge transfer are given by Equation (Equation 16). Hence, the maximum frequency is determined by the maximum difference of eigenenergies, i.e., by the upper and lower limits of the HOMO or LUMO band (calculated for large *N*; cf. Figure 3 and Figure 4). These maximum frequencies for all studied polymers are shown in Figure 8.

The Fourier spectra of the time-dependent probability to find an extra electron or hole at each monomer are generally in the THz regime, mainly in the FIR (far infrared) and MIR (middle infrared) part of the electromagnetic spectrum. When the dominant frequencies, i.e., those with greater Fourier amplitudes, are smaller (larger), the carrier transfer is slower (faster). Extensive examples of the Fourier spectra of the probability to find an extra carrier at the first and at the last monomer, having placed it initially at the first monomer, for I and D aperiodic polymers, for the HOMO and the LUMO regime, can be found in Refs. [85,86]. Recently, we have also analyzed [18,87] the frequency content of periodic polymers, using the TB wire model or the TB extended ladder model, including the Fourier spectra, the WMFs and the TWMF as a function of *N*, with details in Refs. [88,89].

In Figure 9, we depict the TWMF as a function of *N* for the various types of aperiodic polymers. We notice that the TWMF generally stabilizes as the generation number increases. In all cases of *aperiodic* polymers studied in this work, the TWMFs are in the region $\approx 10^{-2}$–$10^{2}$ THz. In various cases of *periodic* I and D polymers studied in Ref. [18], the TWMFs were found in the region $\approx 10^{0}$–$10^{2}$ THz.

### 4.4. *Pure* Mean Transfer Rates

Next, we study the pure mean transfer rates from the first to the last monomer, $k_{1,N}$, or from now on, just *k*. We depict k(N) either for HOMO or for LUMO, for I and D polymers in Figure 10. In all cases, k(N) is a decreasing function. Generally, the degree of coherent transfer difficulty is greater for D polymers. Overall, our results suggest that I polymers, which are simpler cases in terms of energy intricacy, are more efficient regarding coherent hole and electron transfer.

We include in each panel of Figure 10, k(N) of homopolymers (e.g., A…) which are the “champions” among periodic polymers in terms of efficiency of coherent carrier transfer [18], i.e., in terms of magnitude of *k* and of slower decrease of k(N). It seems that k(N) of homopolymers is an unreachable limit for aperiodic polymers. Comparing the *periodic* polymers studied in Refs. [15,16,18] with the *aperiodic* polymers studied in this work, in terms of k(N), we realize that although generally periodic polymers are more efficient, specific aperiodic polymers can be better than specific periodic ones. However, the general picture is that charge transfer in *aperiodic* polymers is orders of magnitude worse than in *periodic* polymers.

In each panel of Figure 10, we also take the best of aperiodic polymers in terms of k(N) and shuffle randomly the sequence of its monomers. In all cases, except for Cantor Set HOMO, this random shuffle deteriorates severely k(N). For Cantor Set, A(T) and T(A) have identical k(N) because the Cantor Set rules for A(T) and T(A) produce equivalent polymers, cf. Equation (Equation 7). For equivalent polymers, k(N) from the first to the last monomer are identical, cf. Equation (Equation 22). For example, TAT ≡ ATA, TATAAATAT ≡ ATATTTATA, TATAAATATAAAAAAAAATATAAATAT ≡ ATATTTATATTTTTTTTTATATTTATA and so on. Similarly, the Cantor Set rules for G(C) and C(G) produce equivalent polymers, which have identical k(N). In Cantor Set HOMO, the best sequences in terms of k(N) are A(T) and T(A), where the hopping integrals involved are $t_{\mathrm{AA}}=t_{\mathrm{TT}}=-$ 8 meV, $t_{\mathrm{AT}}=$ 20 meV, $t_{\mathrm{TA}}=$ 47 meV, and we have just one on-site energy, that of A-T. From these hopping integrals, $t_{\mathrm{AA}}$ has the smallest absolute value. Given the structure of the Cantor Set sequences, making the random shuffle, the number of $t_{\mathrm{AA}}$ decreases, while the numbers of the larger hopping integrals, $t_{\mathrm{AT}}$ and $t_{\mathrm{TA}}$ increase. For this reason, in Cantor Set HOMO, the random shuffle increases k(N). In Cantor Set LUMO, this argument is inverted because now the best sequences in terms of k(N) are G(C) and C(G), where the hopping integrals involved are $t_{\mathrm{GG}}=t_{\mathrm{CC}}=$ 20 meV, $t_{\mathrm{GC}}=-$10 meV, $t_{\mathrm{CG}}=-$8 meV, and we have just one on-site energy, that of G-C. In this case, the random shuffle decreases the number of the larger hopping integrals $t_{\mathrm{GG}}=t_{\mathrm{CC}}$ and increases the number of the smaller hopping integrals $t_{\mathrm{GC}}$ and $t_{\mathrm{CG}}$. However, apart from the exception of the Cantor set HOMO, generally speaking, the conclusion is that deterministic aperiodic polymers possess some kind of order, i.e., a well-defined construction rule that makes them more efficient than random polymers in terms of k(N); therefore, when this rule is destroyed, the transfer efficiency diminishes.

### 4.5. Transfer Rates in Experiments

Comparison of the coherent pure mean transfer rates *k* of our prototype system, B-DNA, with experimentally obtained transfer rates is a rather complicated issue. In the past, the experimental transfer rates in donor—bridge (DNA)—acceptor systems were obtained using the concentrations of different products generated e.g., when a hole is (PY) or is not (PN) transferred. The concentrations of PY and PN were indirectly measured by methods like polyacrylamide gel electrophoresis and piperidine treatment [90,91]. Although these methods revealed some aspects of hole transfer like the sequence dependence and the ability of transfer, they do not provide the kinetics of hole transfer in DNA [92]. Although, generally, greater concentration of PY implies greater charge transfer, there is no proof that the concentrations of PN and PY are proportional to the degree of transfer.

Quantum mechanically, only a fraction of the carrier reaches the acceptor through the bridge. For the same reason, the definition of transfer time is problematic. The transfer rate should depend both on the amount and the speed of transfer. However, the concentration of PY is not strictly proportional to the amount of carrier transfer and not strictly inversely proportional to the time of transfer. A more direct experimental approach is time-resolved spectroscopy, e.g., transient absorption, to observe the products of charge transfer [92,93,94].

Our point of view is different, since the quantity we use, the *pure* mean transfer rate [14], given by Equation (Equation 21), uses simultaneously the magnitude of coherent charge transfer and the time scale of the phenomenon. However, our method applies to coherent transfer only and cannot cover incoherent mechanisms like thermal hopping.

It is a common assertion in the literature that when the fall of the transfer rate with respect to the length of a given DNA segment is described by an exponential fit, the mechanism of transfer is superexchange, whereas when it is described by a power law fit, the mechanism of transfer is multi-step hopping. However, we stress that the fitted parameters produced this way should be treated with care, especially when it comes to attributing them to specific mechanisms. For example, in Ref. [92], where the hole transfer kinetics of various short DNA segments were experimentally investigated with time-resolved spectroscopy, the authors present an exponential decay length β=1.6 Å${}^{-1}$ by fitting the experimental hole transfer rates of G(A)${}_{n}$G DNA oligomers (n=0,1,2) to the exponential law $K=K_{0}e^{-\beta d}$, where *d* is the charge transfer distance, i.e., d=3.4×(N−1) Å. Using the transfer rate values of Ref. [92], we observed that, although β, determined as the slope of the linear fit $\ln\left(K\right)=\ln\left(K_{0}\right)-\beta d$ is indeed ≅1.6 Å${}^{-1}$, a direct exponential fit gives β≅1.3 Å${}^{-1}$, suggesting that the law of decay is not exactly exponential. On the contrary, the fits of our theoretically obtained *pure* mean transfer rates, *k*, for the same system, give β≅1.84 Å${}^{-1}$ for β determined as the slope of the linear fit $\ln\left(k\right)=\ln\left(k_{0}\right)-\beta d$, and β≅1.79 Å${}^{-1}$ for a direct exponential fit $k=k_{0}e^{-\beta d}$, suggesting closer convergence to an exponential decay. Similarly, in Ref. [95], the authors experimentally study, with time-resolved spectroscopy, hole transfer through (GA)${}_{n}$ and (GT)${}_{n}$ sequences, where n= 2–12 is the number of repetition units. The authors fitted the obtained transfer rates to the power law $K=K_{0}^{{}^{\prime }}N^{-\eta }$, where N is the number of hopping steps between guanines (in our notation, $N=\frac{N}{2}-1$), reported the same exponent for both sequences, i.e., η=2, and suggested that this value provides evidence that the long-distance hole transfer occurs by multi-step hopping between guanines. From the rate values provided in Table I of Ref. [95], we observed that, although η as a slope of the linear fit $\ln\left(K\right)=\ln\left(K_{0}^{{}^{\prime }}\right)-\eta \ln\left(N\right)$ is indeed 2 for both sequences, a direct power law fit yields η≅1.4 for (GA)${}_{n}$ and η≅1.3 for (GT)${}_{n}$, suggesting that the rate decay does not follow exactly a power law. On the contrary, the fits of our theoretically obtained *pure* mean transfer rates, *k*, for (GA)${}_{n}$, give η≅1.40 for η determined as the slope of the linear fit $\ln\left(k\right)=\ln\left(k_{0}^{{}^{\prime }}\right)-\eta \ln\left(N\right)$, and η≅1.56 Å${}^{-1}$ for a direct power law fit $k=k_{0}^{{}^{\prime }}N^{-\eta }$. The respective values for (GT)${}_{n}$ are η≅2 for both fits. Hence, our theoretical results suggest that the fall of *k*, as the length of the bridge increases, convergences to a power law and that the fall of the transfer rate is less steep when purines are on the same strand compared to the case when purines are crosswise.

DNA is a dynamical structure, i.e., the geometry is not fixed. Large variations of the TB parameters are expected in real situations and also large variations of the TB parameters have been obtained by different theoretical methods by different authors, cf. e.g., Ref. [14] and references therein. Hence, the parameters any TB model uses have to be utilized with care. In Ref. [96], the authors report experimentally deduced (by transient absorption spectroscopy) charge separation rates, in capped A${}_{n}$ (n= 1–7) and A${}_{3}$G${}_{n}$ (n= 1–19) DNA hairpins with a stilbenedicarboxamide hole donor and a stilbenediether hole acceptor. We computed our theoretical coherent pure mean transfer rates, *k*, for the same systems with a modified parametrization: $t_{AA}\to 1.6t_{AA}$, $t_{AG}\to 2.1t_{AG}$, $t_{GG}\to 2.25t_{AG}$ (cf. Table 2). In order to mimic the donor and the acceptor, we added two sites at the ends of the TB chain, with on-site energies $E_{\mathrm{don}}=E_{A-T}-0.1$ eV, $E_{\mathrm{ac}}=E_{G-C}+0.1$ eV. We used for the hopping integral from the donor (last base pair) to the first base pair (acceptor) 100 meV (250 meV). Our results, along with the experimental ones, are depicted in Figure 11. Apart from the A${}_{1}$ and A${}_{2}$ systems, for which we find much larger rates, the pure mean transfer rates *k* are of the same order of magnitude, in good quantitative agreement with the experimental transfer rates *K*. Actually, the same sequences A${}_{n}$ (n= 1–7) and A${}_{3}$G${}_{n}$ (n= 1–19) analyzed in Ref. [96] had also been analyzed by the same group in Ref. [97]. In Ref. [97], the authors mention a time resolution of ca. 180 fs. Hence, roughly, only transfer rates K<(1/180) PHz ≈(1/200) PHz = 5 × 10${}^{-3}$ PHz can be detected by this technique.

## 5. Conclusions

We systematically studied the energy structure and the coherent transfer of an extra carrier, electron or hole, along various categories of binary quasi-periodic (Fibonacci, Thue–Morse, Double-Period, Rudin–Shapiro) and fractal (Cantor Set, Asymmetric Cantor Set) polymers consisting of either the same monomer (I polymers) or different monomers (D polymers), using the TB wire model and B-DNA as a prototype system.

Regarding the energy structure of the polymers, we calculated the HOMO and LUMO eigenspectra and the density of states, as well as the HOMO–LUMO gap. The eigenenergies lie around the monomers’ on-site energies. We demonstrated numerically and proved analytically that for I polymers, the eigenenergies are always symmetric relative to the (constant) monomer on-site energy. For both I and D polymers, in quasi-periodic cases, the DOS has rather acute subbands, while in fractal cases, it is fragmented and spiky. D polymers pose smaller HOMO–LUMO gaps than I polymers. The G-C (A-T) monomer gap is always greater than the gaps of I polymers made of G and C or A and T. The lower HOMO (LUMO) band limit of D polymers is always between the lower and upper HOMO (LUMO) band limit of I polymers consisted of A and T, while the upper HOMO (LUMO) band limit of D polymers is always between the lower and upper HOMO (LUMO) band limit of I polymers consisted of G and C.

Next, we studied the mean over time probabilities to find an extra hole or electron at each monomer of the polymer, having it initially placed at the first monomer. For I polymers, the mean over time probabilities are significant only rather close to the first monomer, although in some cases we observe non-negligible probabilities at more distant monomers. For D polymers, the mean over time probabilities are generally negligible further than the first monomer. This situation in *aperiodic* polymers, where the mean over time probabilities generally decline a little away from the site where the carrier was initially placed, is in contrast to the situation in *periodic* polymers studied e.g., in Refs. [15,16,18], where, generally, non-negligible probabilities exist at distant—from the initial—sites.

Furthermore, we determined the frequency content of coherent extra carrier transfer via the total weighted mean frequency of the polymer, using the weighted mean frequencies of the Fourier spectra that correspond to the probabilities to find the carrier at each monomer. The TWMF generally stabilizes after a few generations. We showed that, in all cases of *aperiodic* polymers studied in this work, the TWMF lies in the regime $\approx 10^{-2}-10^{2}$ THz. This is different from various cases of *periodic* I and D polymers studied in Ref. [18], where the TWMFs were found in the region $\approx 10^{0}-10^{2}$ THz.

The study of the pure mean transfer rates, k(N), shows that I polymers, which are simpler cases in terms of energy intricacy, are more efficient than D polymers regarding coherent hole and electron transfer. Comparing the *periodic* polymers studied in Refs. [15,16,18] with the *aperiodic* polymers studied in this work, in terms of k(N), we realize that, although generally periodic polymers are more efficient, specific aperiodic polymers can be better than specific periodic ones. However, the general picture is that charge transfer in *aperiodic* polymers is orders of magnitude worse than in *periodic* polymers. The structurally simplest periodic polymers, i.e., the homopolymers [18], represent an unreachable limit for all aperiodic polymers. Furthermore, a random shuffle of a quasi-periodic or fractal monomer sequence destroys the deterministic character of its construction rules, thus leading to vanishing transfer rates. As far as comparison with experiments is concerned, large variations of the TB parameters are expected in real situations, hence modifications are necessary. Using a modified parametrization, we were able to find hole pure mean transfer rates *k* of similar magnitude with experimental transfer rates *K* obtained by time-resolved spectroscopy.

## Figures and Tables

**Figure 1 materials-12-02177-f001:**
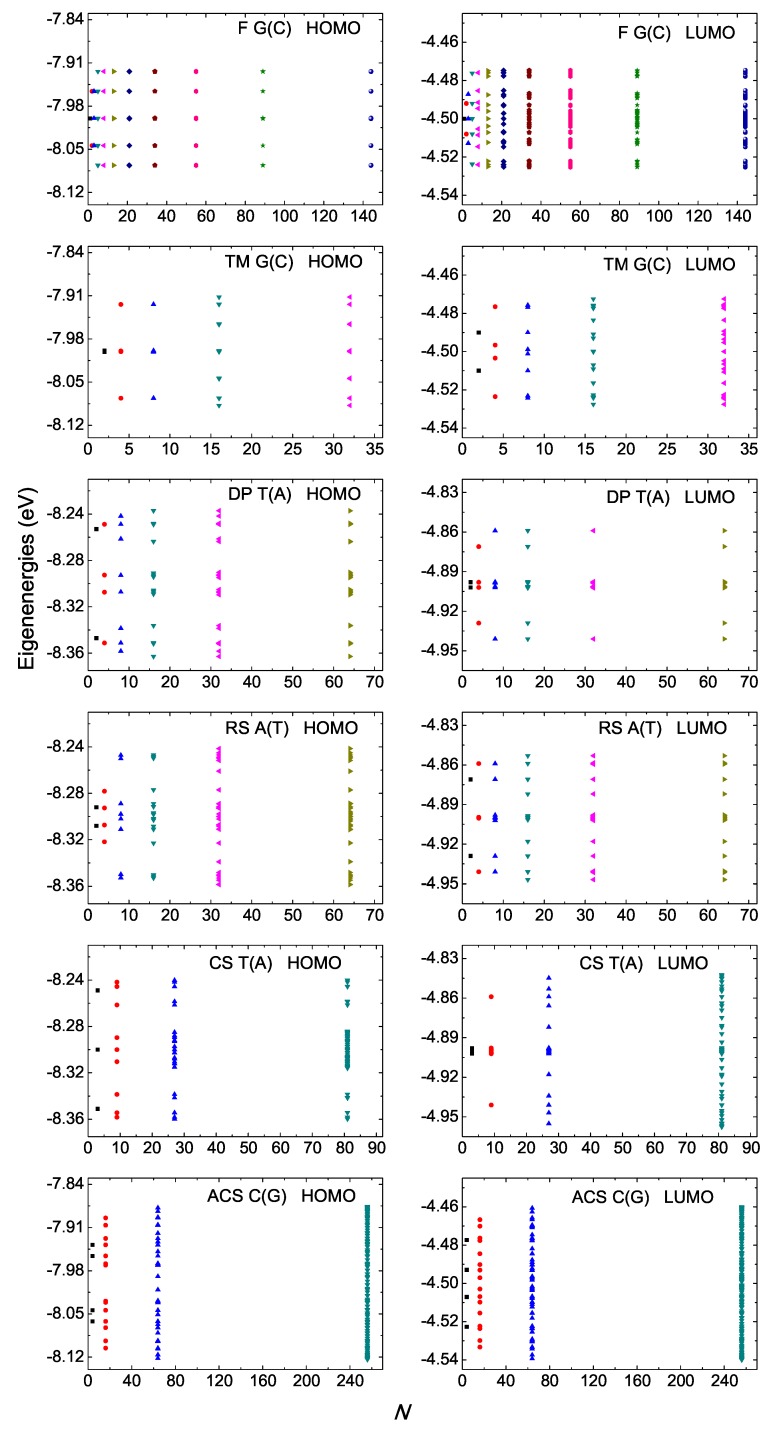
Eigenspectra of F G(C), TM G(C), DP T(A), RS A(T), CS T(A), ACS C(G) polymers, for the HOMO (left column) regime and the LUMO (right column) regime, for a few generations. The horizontal axis shows the number of monomers in the polymer *N*. Different colors correspond to different generations. The notation of polymers is given in Table 1.

**Figure 2 materials-12-02177-f002:**
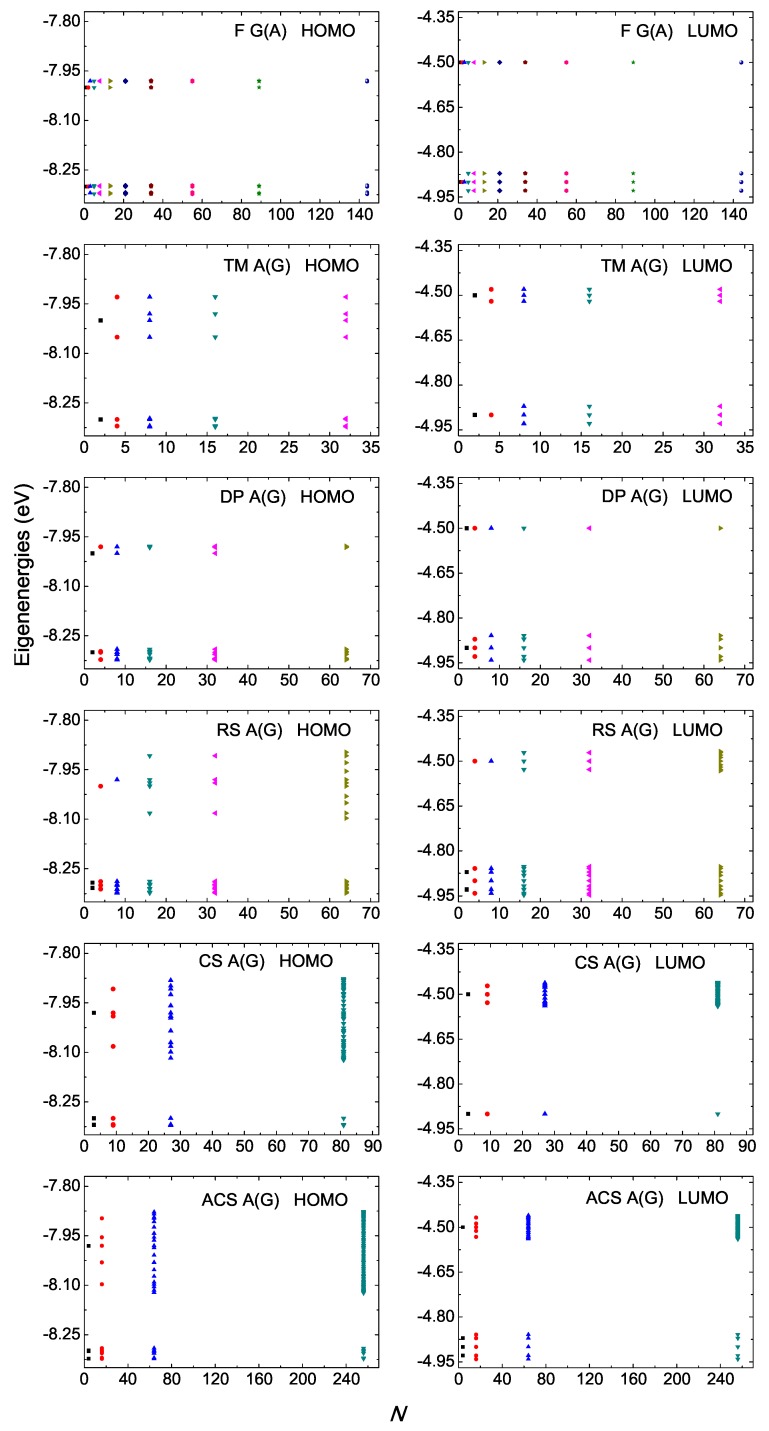
Eigenspectra of F A(G), TM A(G), DP A(G), RS A(G), CS A(G), ACS A(G) polymers, for the HOMO (left column) regime and the LUMO (right column) regime, for a few generations. The horizontal axis shows the number of monomers in the polymer *N*. Different colors correspond to different generations. The notation of polymers is given in Table 1.

**Figure 3 materials-12-02177-f003:**
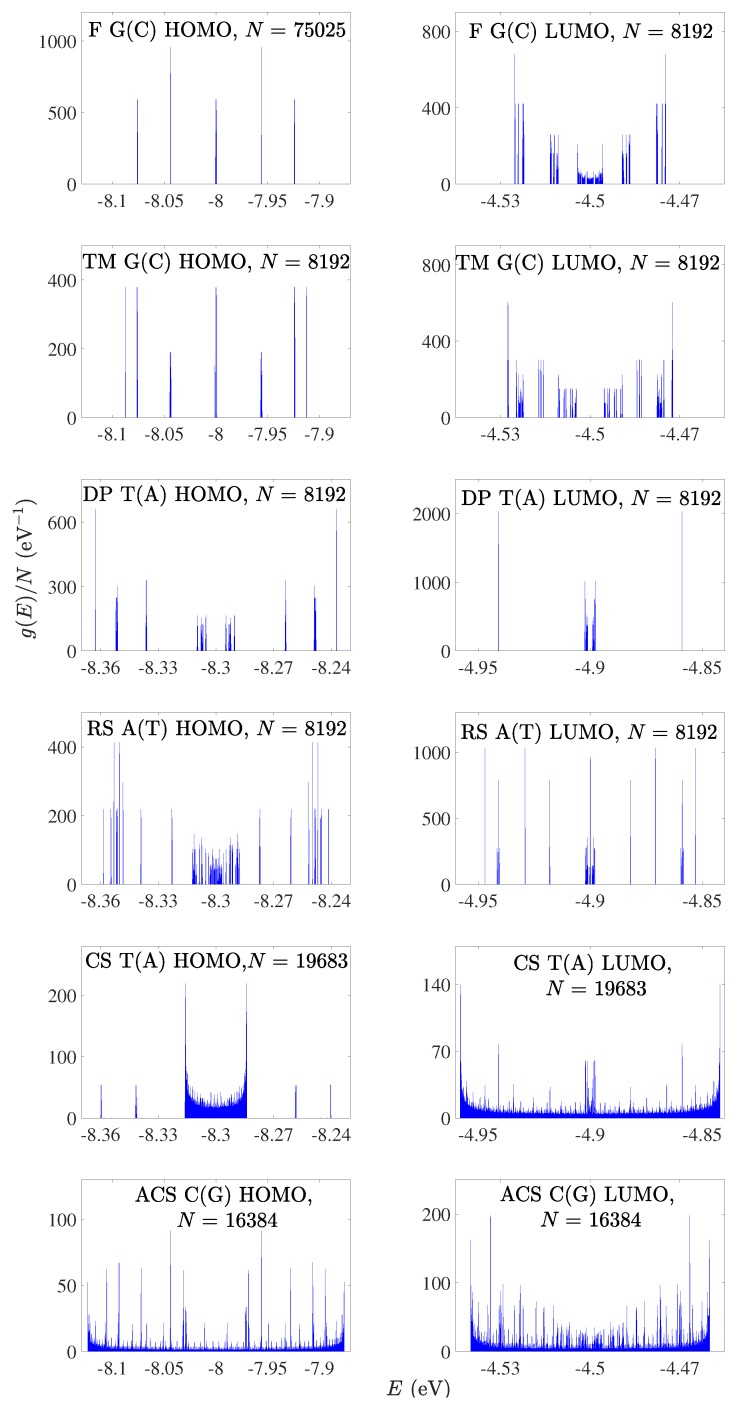
Density of states of F G(C), TM G(C), DP T(A), RS A(T), CS T(A), ACS C(G) polymers, for the HOMO (left column) regime and the LUMO (right column) regime, for a generation with large *N*. The notation of polymers is given in Table 1.

**Figure 4 materials-12-02177-f004:**
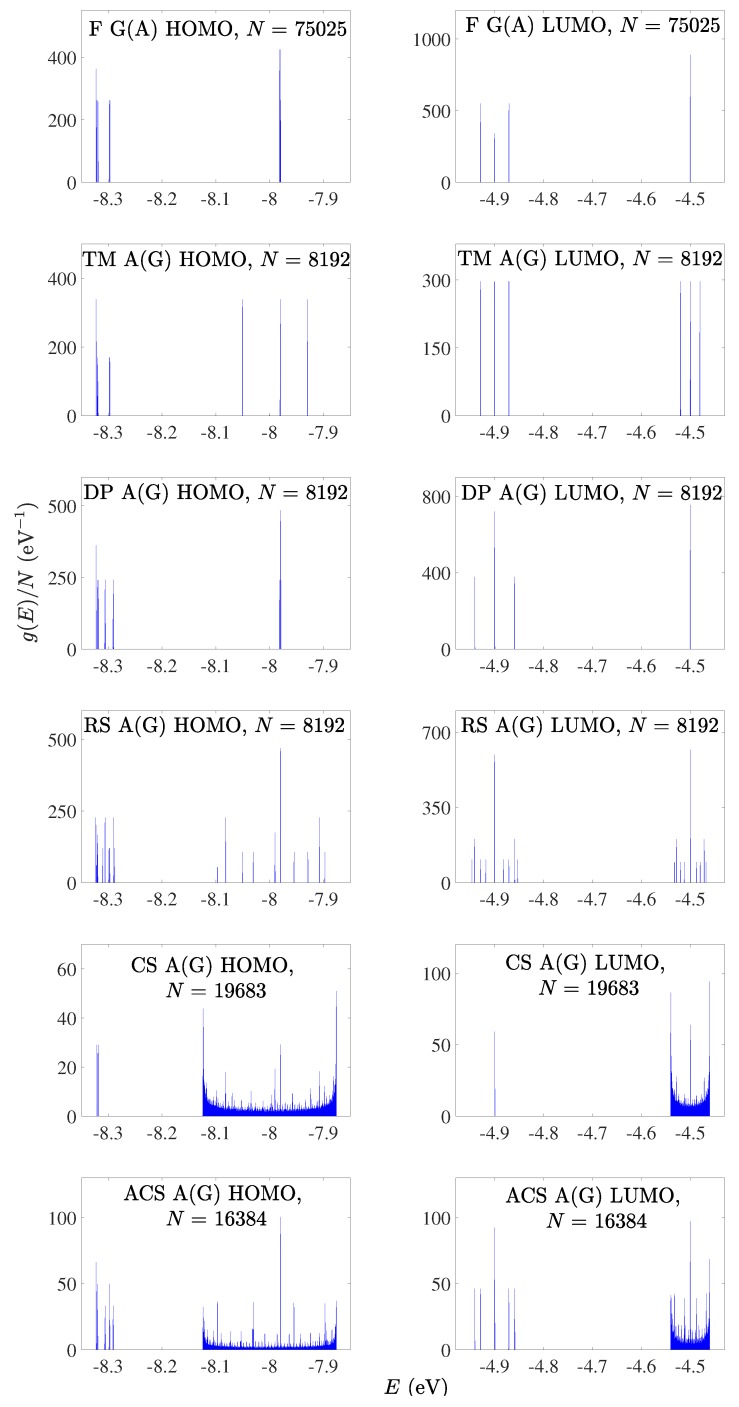
Density of states of F G(A), TM A(G), DP A(G), RS A(G), CS A(G), ACS A(G) polymers, for the HOMO (left column) regime and the LUMO (right column) regime, for a generation with large *N*. The notation of polymers is given in Table 1.

**Figure 5 materials-12-02177-f005:**
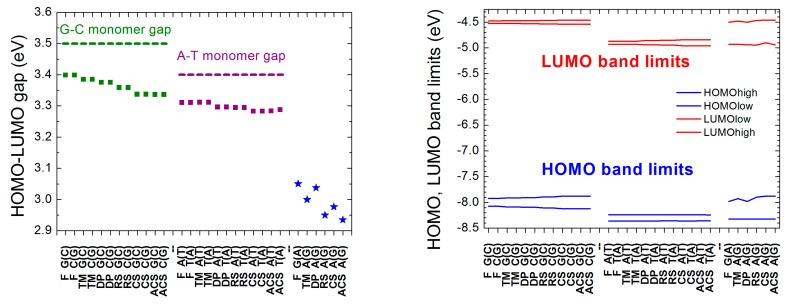
Energy gaps (**left**) as well as HOMO and LUMO band limits (**right**), at the large *N* limit, for all aperiodic polymers considered in this work. Squares: I Polymers, i.e., made of the same monomer. Blue stars: D Polymers, i.e., made of different monomers. The green (purple) dashed line shows the energy gap of the G-C (A-T) base pair. The notation of polymers is given in Table 1.

**Figure 6 materials-12-02177-f006:**
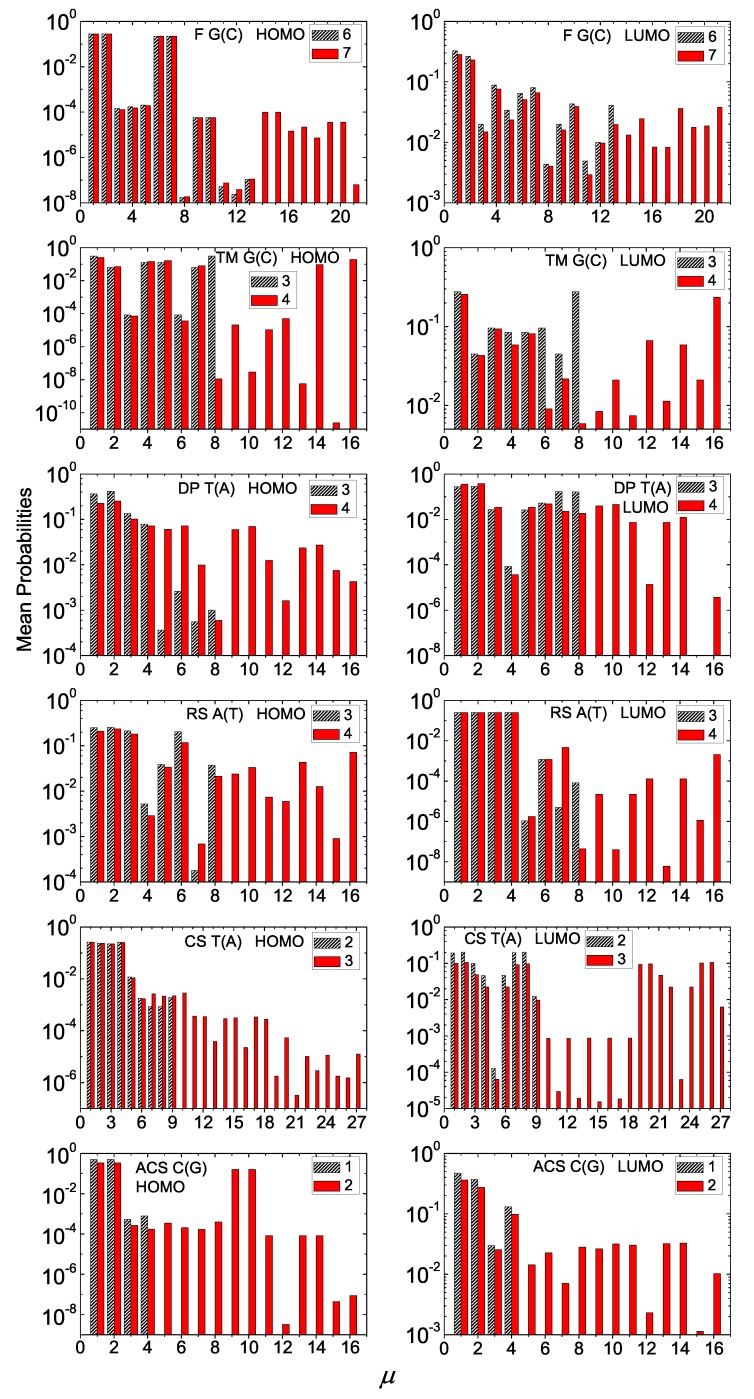
Mean over time probabilities to find the extra carrier at each monomer μ=1,…,N, having placed it initially at the first monomer, for two consecutive generations (the number of which is denoted at each panel’s legend) for G(C), TM G(C), DP T(A), RS A(T), CS T(A), ACS C(G) polymers, for HOMO (left column) and LUMO (right column). The notation of polymers is given in Table 1.

**Figure 7 materials-12-02177-f007:**
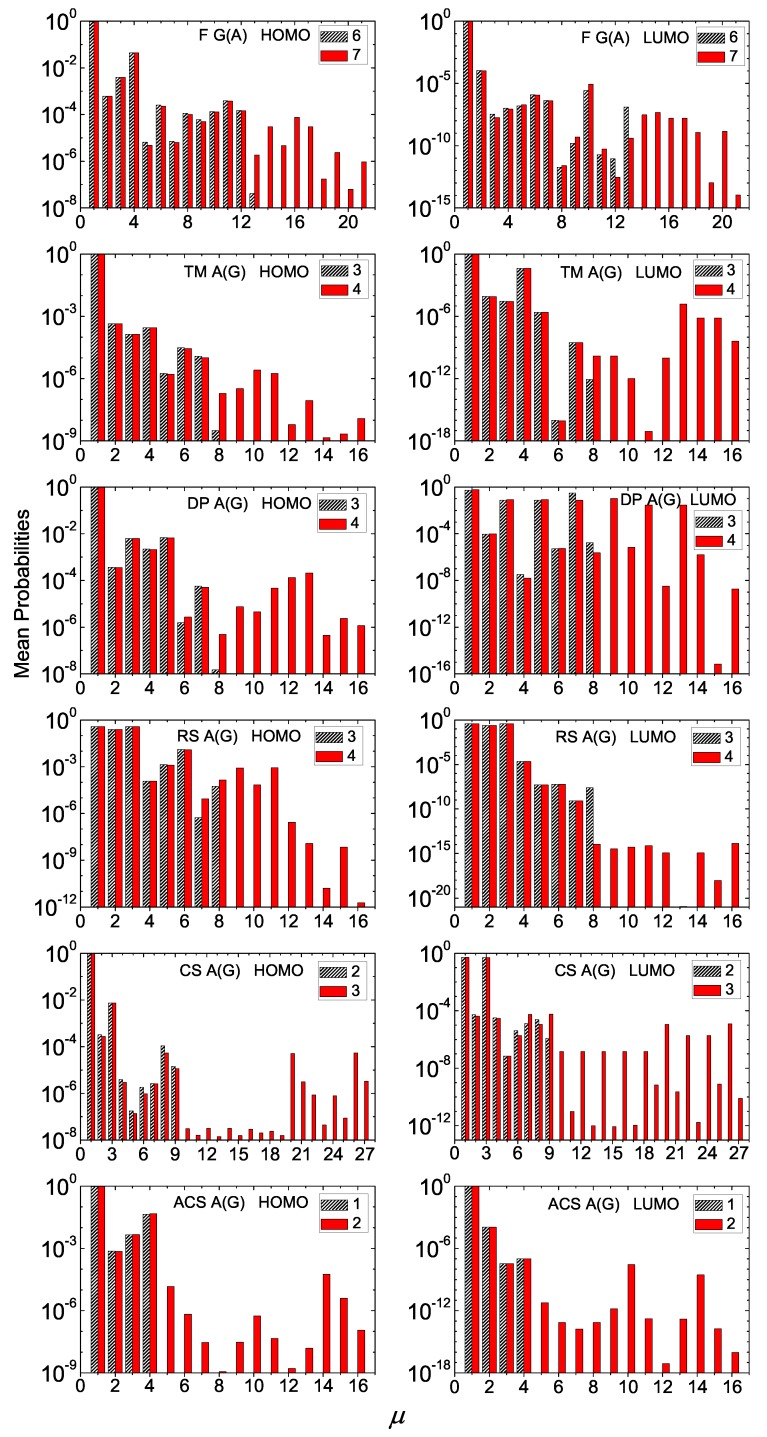
Mean over time probabilities to find the extra carrier at each monomer μ=1,…,N, having placed it initially at the first monomer, for two consecutive generations (the number of which is denoted at each panel’s legend) for F G(A), TM A(G), DP A(G), RS A(G), CS A(G), ACS A(G) polymers, for HOMO (left column) and LUMO (right column). The notation of polymers is given in Table 1.

**Figure 8 materials-12-02177-f008:**
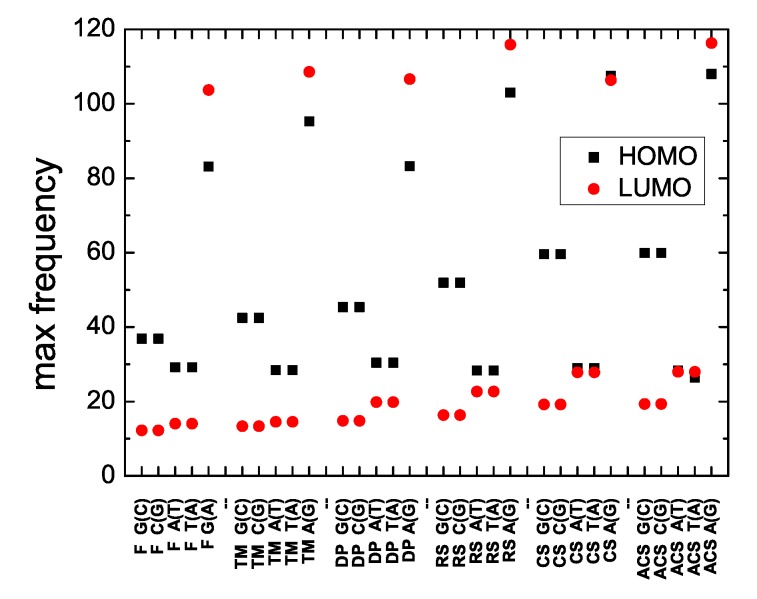
The maximum frequency of the Fourier spectrum, for the HOMO and the LUMO regime of Fibonacci, Thue–Morse, Double Period, Rudin–Shapiro, Cantor Set, Asymmetric Cantor Set polymers, at the large *N* limit. The notation of polymers is given in Table 1.

**Figure 9 materials-12-02177-f009:**
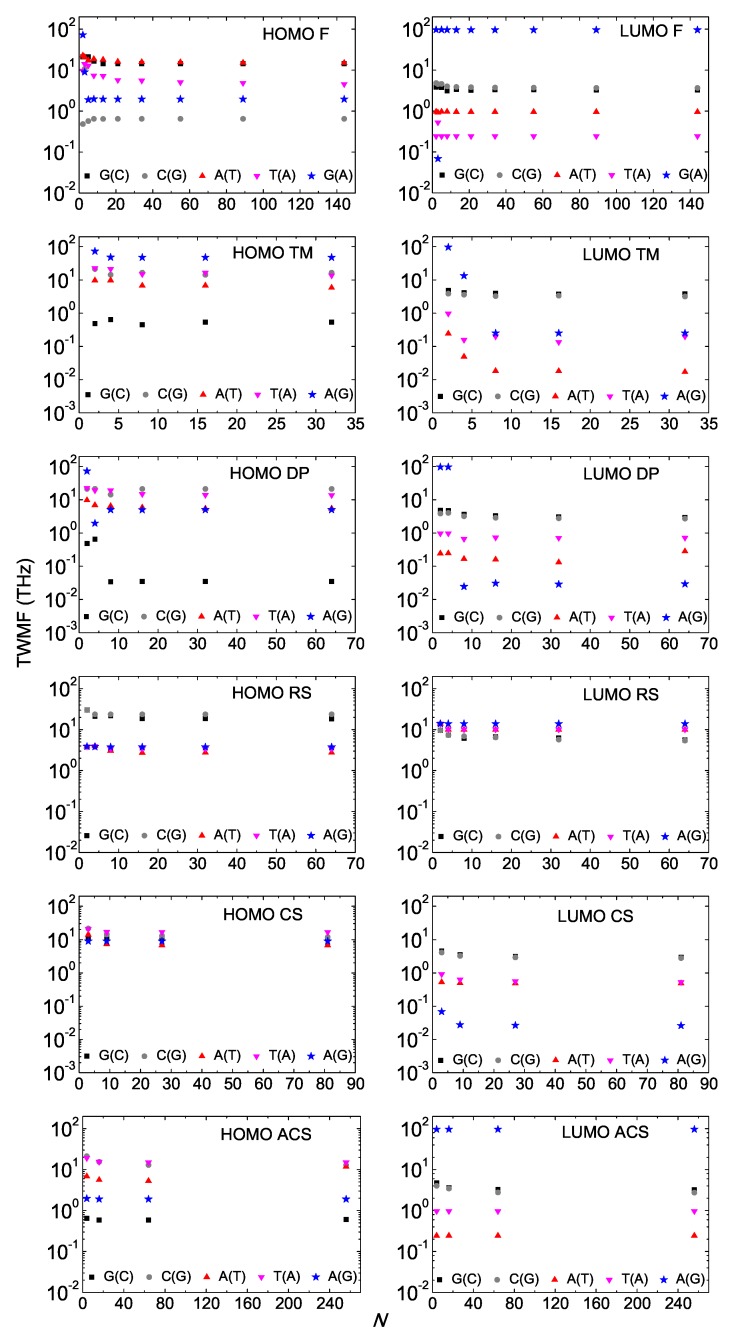
Total Weighted Mean Frequency (TWMF) as a function of the number of monomers *N* in the polymer, having placed the carrier initially at the first monomer, for Fibonacci, Double Period, Rudin–Shapiro, Cantor Set, Asymmetric Cantor Set polymers, for the HOMO (left column) regime and the LUMO (right column) regime. D Polymers, i.e., made of different monomers, are denoted by blue stars. The notation of polymers is given in Table 1.

**Figure 10 materials-12-02177-f010:**
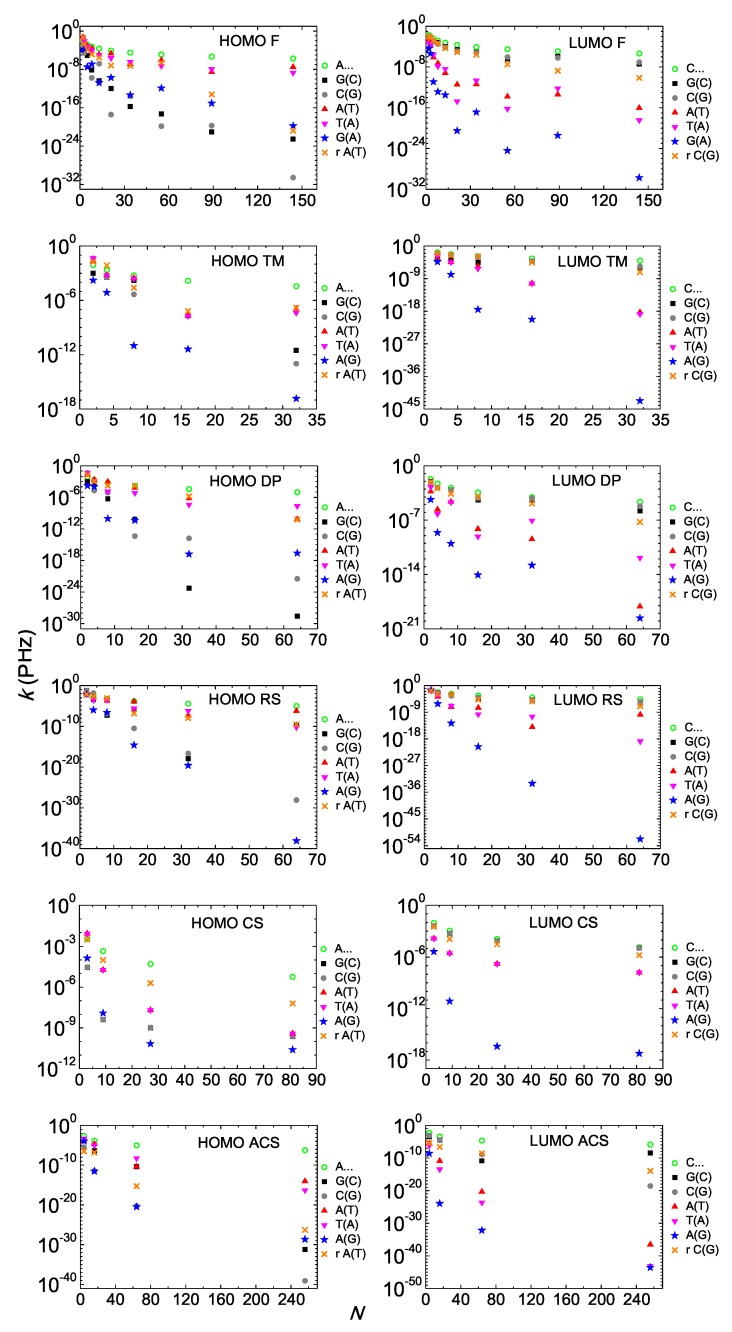
*Pure* mean transfer rates *k* of Fibonacci, Thue–Morse, Double Period, Rudin–Shapiro, Cantor Set, Asymmetric Cantor Set polymers, homopolymers and randomly shuffled aperiodic polymers as a function of the number of monomers *N* in the polymer, for the HOMO (left column) regime and the LUMO (right column) regime. The blue stars denote the D Polymers, i.e., made of different monomers. The notation of polymers is given in Table 1.

**Figure 11 materials-12-02177-f011:**
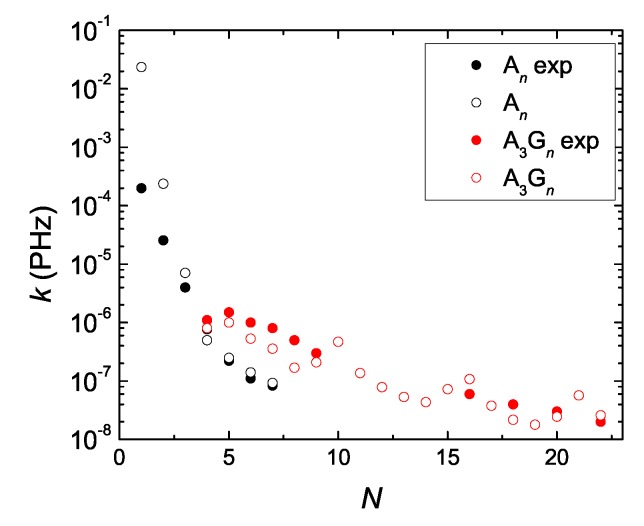
Comparison of experimental hole transfer rates *K* for A${}_{n}$ and A${}_{3}$G${}_{n}$ segments [96] (full circles) with our theoretical coherent pure mean transfer rates *k* (empty circles), as a function of the number of monomers *N* in the polymer. The Tight-Binding parametrization is described in the main text.

**Table 1 materials-12-02177-t001:** Examples of the types of polymers studied in this work. I (D) denotes polymers made of identical (different) monomers. We only mention the $5^{\prime }$-$3^{\prime }$ base sequence along one of the two strands. X(Y) means that the zeroth generation (the “seed”) of the binary sequence contains solely X and the other letter in the sequence is Y.

Type	Sequence Example	Notation
Fibonacci I	G, C, CG, CGC, CGCCG, …	F G(C)
Fibonacci D	G, A, AG, AGA, AGAAG, …	F G(A)
Thue–Morse I	G, GC, GCCG, GCCGCGGC, …	TM G(C)
Thue–Morse D	A, AG, AGGA, AGGAGAAG, …	TM A(G)
Double Period I	T, TA, TATT, TATTTATA, …	DP T(A)
Double Period D	A, AG, AGAA, AGAAAGAG, …	DP A(G)
Rudin–Shapiro I	AA, AAAT, AAATAATA, …	RS A(T)
Rudin–Shapiro D	AA, AAAG, AAAGAAGA, …	RS A(G)
Cantor Set I	T, TAT, TATAAATAT, …	CS T(A)
Cantor Set D	A, AGA, AGAGGGAGA, …	CS A(G)
Asymmetric Cantor Set I	C, CGCC, CGCCGGGGCGCCCGCC, …	ACS C(G)
Asymmetric Cantor Set D	A, AGAA, AGAAGGGGAGAAAGAA, …	ACS A(G)

**Table 2 materials-12-02177-t002:** The HOMO/LUMO hopping integrals $t_{\mu ,\lambda }$, in meV, between successive base pairs μ,λ.

μ,λ	$t_{\mu ,\lambda }$H	$t_{\mu ,\lambda }$L Ref. [19]	μ,λ Ref. [19]	$t_{\mu ,\lambda }$H Ref. [19]	$t_{\mu ,\lambda }$L Ref. [19]
AA ≡ TT	−8	−29	AT	20	0.5
AG ≡ CT	−5	3	AC ≡ GT	2	32
TA	47	2	TG ≡ CA	−4	17
TC ≡ GA	−79	−1	GG ≡ CC	−62	20
GC	1	−10	CG	−44	−8

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
