# Peer review of "Quasi-Periodic and Fractal Polymers: Energy Structure and Carrier Transfer"

_materials, 2019, doi:10.3390/ma12132177_

Reviewer 1 Report

The manuscript deals with the energy structure and the coherent transfer of an extra electron or hole along 1aperiodic polymers made of N monomers, with fixed boundaries, using B-DNA as a prototype 2
system. Authors indicated that the experimental results on charge transfer rates in 14DNA are in accordance with their coherent pure mean transfer rates.
The manuscript responds to the objectives defined in the title and developed in the introduction, the results are in accordance with the advanced explanation in the discussion part.
After a careful proof-reading of the manuscript, this research article is recommended for publication in Materials with some remarks and suggestions.
- Please include the sources of all used experimental techniques.
- Please be more precise about the reproducibility and repeatability of each technique used.
- The statistical analysis is not really explained. Please be more precise, show results and discuss them.
- References should be checked as some references are not properly formatted.

Author Response

Comments and Suggestions for Authors

The manuscript deals with the energy structure and the coherent transfer of an extra electron or hole along aperiodic polymers made of N monomers, with fixed boundaries, using B-DNA as a prototype system. Authors indicated that the experimental results on charge transfer rates in DNA are in accordance with their coherent pure mean transfer rates. The manuscript responds to the objectives defined in the title and developed in the introduction, the results are in accordance with the advanced explanation in the discussion part. After a careful proof-reading of the manuscript, this research article is recommended for publication in Materials with some remarks and suggestions.

We thank the referee for his/her time and effort as well as for recommending its publication in Materials with minor revisions. Below we reply to his/her suggestions:
- Please include the sources of all used experimental techniques.

This is a theoretical work. We include a comparison with experiments performed by others in subsection "4.5. Transfer rates in experiments" which covers 2 pages and contains extensive discussion. The references are given, those works have been published in respected scientific journals with referees, so there is not much that we can add. We compare our coherent pure mean transfer rate k with experimentally obtained transfer rates K. Our method applies to coherent transfer only and cannot cover incoherent mechanisms like thermal hopping. The experimental techniques with which we compare our results include methods like polyacrylamide gel electrophoresis and piperidine treatment, references [90,91],  and time-resolved spectroscopy, e.g. transient absorption, references [92–97].

- Please be more precise about the reproducibility and repeatability of each technique used.

The reproducibility and repeatability of these techniques is also a matter for which the original authors should be asked and not us. However, these experimental results have been published in respected scientific journals with referees, so there is not much that we can add.

- The statistical analysis is not really explained. Please be more precise, show results and discuss them.

The analysis of our results is done with the equations included in section "3. Theory". Extensive analysis of our fitting methods can be found in references [14,15,16] and in the supplementary material of reference [18]. The article is already 30 pages and includes almost 100 references.

In line with the referee's suggestion, we now include a few phrases at the end of Introduction:

"The analysis of our results is done with the equations included in Sec.~\ref{sec:theory}. Extensive analysis of our fitting methods can be found in Refs.~\cite{Simserides:2014,LChMKTS:2015,LChMKLTTS:2016} and in the Supplemental Material of Ref.~\cite{LVBMS:2018}."

- References should be checked as some references are not properly formatted.

We thank the referee for this observation. We have checked the references 1-1 and modified them in some cases, usually relative to the doi and url format.

Reviewer 2 Report

This is an interesting paper investigating energy structure and carrier transfer process by using a Tight-Binding wire model. Based on the polymers consisting with different monomers, the author calculates the HOMO, LOMO and DOS of the proposed polymers and also identifies the HOMO-LOMO gaps. Although the TB model still needs to be modified for more accurate simulations and predictions, this study provides respectable knowledge in this field. I suggest to publish this work with minor revision. More comments are attached:

1.       In the introduction section, “…. while, the term transport implies the application of voltage between electrodes.” I can’t understand about this sentence. Can the authors explain more about this? Or are there any references?

2.       Line 65& 66, the HOMO should be defined where is first appear

3.       Figure 1 & 2, what the different colors stand for, or just different numbers of monomers?

Author Response

Comments and Suggestions for Authors

This is an interesting paper investigating energy structure and carrier transfer process by using a Tight-Binding wire model. Based on the polymers consisting with different monomers, the author calculates the HOMO, LOMO and DOS of the proposed polymers and also identifies the HOMO-LOMO gaps. Although the TB model still needs to be modified for more accurate simulations and predictions, this study provides respectable knowledge in this field. I suggest to publish this work with minor revision.

We thank the referee for his/her time and effort as well as for recommending its publication in Materials with minor revisions. Below we reply to his/her suggestions:

More comments are attached:

1.       In the introduction section, “…. while, the term transport implies the application of voltage between electrodes.” I can’t understand about this sentence. Can the authors explain more about this? Or are there any references?

Although it is not rare to see or hear the terms "transport" and "transfer" indiscriminately, if we want to be more precise, "transport" means that the system under investigation is held between electrodes and that a voltage is applied between these electrodes, while, the term "transfer" means that a carrier, created (e.g. by oxidation or reduction) or injected at a specific place, moves to a more favorable location, without the application of external voltage.

So, we have changed the piece of text:

"The term \textit{transfer} means that a carrier, created (e.g. by oxidation or reduction) or injected at a specific place, moves to a more favorable location, while, the term \textit{transport} implies the application of voltage between electrodes."

with the piece of text:

"Although it is  not rare to see or hear the terms transport and transfer indiscriminately, if we want to be more precise, transport means that the system under investigation is held between electrodes and that a voltage is applied between these electrodes, while, the term transfer means that a carrier, created (e.g. by oxidation or reduction) or injected at a specific place, moves to a more favorable location, without the application of external voltage."

2.       Line 65& 66, the HOMO should be defined where is first appear

We thank the referee for this observation. We removed the explanation of the HOMO abbreviation from that point. Now we include the explanations of HOMO and LUMO in the abstract: "highest occupied molecular orbital (HOMO) eigenspectrum and the lowest unoccupied molecular orbital (LUMO) eigenspectrum" as well as in their first appearances in the main text: "In natural DNA, it is more likely that a hole will be created at a guanine which has the highest HOMO (highest occupied molecular orbital) of all bases~\cite{HKS:2010-2011} and an electron will be created at a thymine which has the lowest LUMO (lowest unoccupied molecular orbital) of all bases~\cite{HKS:2010-2011}."

3.       Figure 1 & 2, what the different colors stand for, or just different numbers of monomers?

We thank the referee for this observation. Different colors stand for different generations. For example, in the Fibonacci sequence, the number of monomers in the polymer N, takes the

following values from generation to generation: 1,1,2,3,5,8,13,21,34,55,89,144,... Of course, 1,

1, are not included since these are just monomers (N = 1).

In the captions of Figures 1 and 2 we have now added: "Different colors correspond to different generations.".

Also, in the beginning of Subsection "Eigenspectra, Density of States, Energy Gaps", we have now added: "In Figs. 1 and 2,

different colors correspond to different generations, e.g., in the Fibonacci sequence, the number of monomers in the polymer,

N, takes the following values from generation to generation  (1, 1,) 2, 3, 5, 8, 13, 21, 34, 55, 89, 144,... ."

Comments and Suggestions for Authors

This is an interesting paper investigating energy structure and carrier transfer process by using a Tight-Binding wire model. Based on the polymers consisting with different monomers, the author calculates the HOMO, LOMO and DOS of the proposed polymers and also identifies the HOMO-LOMO gaps. Although the TB model still needs to be modified for more accurate simulations and predictions, this study provides respectable knowledge in this field. I suggest to publish this work with minor revision.

We thank the referee for his/her time and effort as well as for recommending its publication in Materials with minor revisions. Below we reply to his/her suggestions:

More comments are attached:

1.       In the introduction section, “…. while, the term transport implies the application of voltage between electrodes.” I can’t understand about this sentence. Can the authors explain more about this? Or are there any references?

Although it is not rare to see or hear the terms "transport" and "transfer" indiscriminately, if we want to be more precise, "transport" means that the system under investigation is held between electrodes and that a voltage is applied between these electrodes, while, the term "transfer" means that a carrier, created (e.g. by oxidation or reduction) or injected at a specific place, moves to a more favorable location, without the application of external voltage.

So, we have changed the piece of text:

"The term \textit{transfer} means that a carrier, created (e.g. by oxidation or reduction) or injected at a specific place, moves to a more favorable location, while, the term \textit{transport} implies the application of voltage between electrodes."

with the piece of text:

"Although it is  not rare to see or hear the terms transport and transfer indiscriminately, if we want to be more precise, transport means that the system under investigation is held between electrodes and that a voltage is applied between these electrodes, while, the term transfer means that a carrier, created (e.g. by oxidation or reduction) or injected at a specific place, moves to a more favorable location, without the application of external voltage."

2.       Line 65& 66, the HOMO should be defined where is first appear

We thank the referee for this observation. We removed the explanation of the HOMO abbreviation from that point. Now we include the explanations of HOMO and LUMO in the abstract: "highest occupied molecular orbital (HOMO) eigenspectrum and the lowest unoccupied molecular orbital (LUMO) eigenspectrum" as well as in their first appearances in the main text: "In natural DNA, it is more likely that a hole will be created at a guanine which has the highest HOMO (highest occupied molecular orbital) of all bases~\cite{HKS:2010-2011} and an electron will be created at a thymine which has the lowest LUMO (lowest unoccupied molecular orbital) of all bases~\cite{HKS:2010-2011}."

3.       Figure 1 & 2, what the different colors stand for, or just different numbers of monomers?

We thank the referee for this observation. Different colors stand for different generations. For example, in the Fibonacci sequence, the number of monomers in the polymer N, takes the following values from generation to generation: 1,1,2,3,5,8,13,21,34,55,89,144,... Of course, 1, 1, are

not included since these are just monomers (N = 1).

In the captions of Figures 1 and 2 we have now added: "Different colors correspond to different generations.".

Also, in the beginning of Subsection "Eigenspectra, Density of States, Energy Gaps", we have now added: "In Figs. 1 and 2,

different colors correspond to different generations, e.g.,

 in the Fibonacci sequence, the number of monomers in the polymer, N,

takes the following values from generation to generation  (1, 1,) 2, 3, 5, 8, 13, 21, 34, 55, 89, 144,... ."
